# Mechanisms of Cellular Responses of the Natural Alkaloid Caulerpin and Its Similarities with the Lipid-Lowering Agent Fenofibrate in *Mytilus galloprovincialis*

**DOI:** 10.3390/toxins17100512

**Published:** 2025-10-18

**Authors:** Michela Panni, Marica Mezzelani, Maria Elisa Giuliani, Paola Nisi Cerioni, Alessandro Nardi, Ernesto Mollo, Francesco Regoli, Maura Benedetti, Stefania Gorbi

**Affiliations:** 1Dipartimento di Scienze della Vita e dell’Ambiente, Università Politecnica delle Marche, Via Brecce Bianche, 60131 Ancona, Italy; m.panni@staff.univpm.it (M.P.); m.mezzelani@univpm.it (M.M.); p.nisi@univpm.it (P.N.C.); a.nardi@univpm.it (A.N.); f.regoli@univpm.it (F.R.); 2Advanced Technology Center for Aging Research and Geriatric Mouse Clinic (IRCCS-INRCA), 60121 Ancona, Italy; m.giuliani@inrca.it; 3National Biodiversity Future Center (NBFC), 90133 Palermo, Italy; 4Institute of Biomolecular Chemistry, National Research Council, 80078 Pozzuoli, Italy; emollo@icb.cnr.it

**Keywords:** caulerpin, fenofibrate, mechanisms of action, *Mytilus galloprovincialis*, lipid metabolism, redox homeostasis, molecular responses

## Abstract

Marine-derived natural products have attracted increasing attention due to their promising pharmacological potential against various diseases. The present study investigated the hypolipidemic properties of the alkaloid caulerpin, a secondary metabolite of green algae of the genus *Caulerpa*, through an ex vivo approach with Precision-Cut Tissue Slices (PCTSs) of *Mytilus galloprovincialis* digestive glands. PCTSs were exposed to caulerpin (CAU) (100 µM) and fenofibrate (FFB) (100 µM) for 24, 48, and 72 h. Analyses of molecular and cellular responses pertaining to lipid metabolism suggested a similar mechanism of action between CAU and FFB in marine invertebrate species, resulting in a decrease in neutral lipid content ranging from 50 to 70%. CAU’s hypolipidemic action was not associated with increased prooxidant conditions, and slower metabolism of the natural alkaloid compared to FFB was indicated by the response of biotransformation and excretion pathways. Overall, these findings provide initial insights into the potential valorization of CAU for pharmaceutical and nutraceutical applications, highlighting the need for further investigation into its mechanisms of action, possible side effects, and interactions with other drugs.

## 1. Introduction

During the last century, advances in medical science have enabled the treatment of several pathologies, leading to a general improvement in health quality and an increase in life expectancy, which has ultimately contributed to the current demographic trend of an aging population [1]. In this context, marine-derived compounds can serve as new active agents for improving human health. Indeed, several marine natural products have recently demonstrated efficacy against a wide range of diseases, owing to their antibacterial, antifungal, antiprotozoal, antituberculosis, antiviral, antidiabetic, anti-inflammatory, and anticancer properties [2,3,4]. Cardiovascular diseases (CVDs) represent one of the greatest health burdens of the modern era, and elevated plasma lipid levels have been identified as the most predictive risk factor for the development and progression of CVDs [5]. For this reason, β-blockers, ACE inhibitors, angiotensin II receptor blockers, and calcium channel blockers are commonly prescribed alongside lipid-lowering agents, such as statins and fibrates, which indirectly reduce the risk of severe CVDs by treating dyslipidemia. Among lipid-lowering agents, fenofibrate (FFB) is an amphipathic carboxylic fibrate with many blood lipid-modifying properties, including decreasing triglyceride levels and increasing high-density lipoprotein (HDL) cholesterol levels. These effects are mediated by the activation of the peroxisome proliferator-activated receptor, PPARα, which enhances peroxisomal β-oxidation and activates lipoprotein lipase [6].

In marine environments, the bisindole alkaloid caulerpin (CAU) is a secondary metabolite of several green algae of the genus *Caulerpa* [7] and other seaweeds [8,9]. Diverse bioactive properties have been attributed to CAU, including antiviral [10,11,12], antituberculosis [13], anti-nociceptive [14], anti-inflammatory [15,16,17], and anticancer [18,19] activities. In addition, growing evidence has driven the hypothesis of potential CAU-mediated effects on lipid metabolism. The first observations of fish feeding on *Caulerpa cylindracea* highlighted that the hepatic accumulation of CAU was associated with the increase in parameters involved in fatty acid β-oxidation, e.g., the activity of acyl CoA oxidase (ACOX) and peroxisome proliferator-activated receptor α (PPARα) transcription [20]. Moreover, a reduction in polyunsaturated fatty acids was observed in the muscle of the same fish [21]. CAU was then identified as the direct cause of alterations in the lipid composition of fish meat [22]. Additionally, in silico, in vitro, and ex vivo experiments demonstrated that this active principle acts as an agonist of PPARs [23]. From this perspective, CAU shows several similarities to FFB [24], and numerous studies have demonstrated FFB’s ability to induce lipid metabolism abnormalities and decrease the content of long-chain omega-3s, such as eicosapentaenoic acid (EPA) and docosahexaenoic acid (DHA), when administered to freshwater fish [25,26]. Notably, the same effects have been observed in wild populations of *D. sargus* feeding on *C. cylindracea* [21].

As mentioned above, hypolipidemic agents like FFB activate lipolysis through the modulation of peroxisomal genes/enzymes: reactive oxygen species generated by peroxisomal reactions also participate in lipolysis [27,28,29,30,31]. For this reason, the activation of lipid metabolism is typically associated with an imbalance of the cellular redox homeostasis that, in turn, can increase oxidative stress [32,33,34,35,36,37,38,39].

In this respect, the present study aimed to investigate the potential of CAU as a marine-derived hypolipidemic candidate in non-target marine species. An ex vivo approach was applied using Precision-Cut Tissue Slices (PCTSs) of the digestive glands of *Mytilus galloprovincialis*, the selected model invertebrate marine species. PCTSs offer the main advantages of cell cultures while preserving the physiological architecture of the tissue. This technique has already been effectively employed to demonstrate the activation of lipid metabolism genes via CAU in the marine fish *Diplodus sargus* [23] and has also been validated for the development of ecotoxicological tests in invertebrate marine organisms like the Mediterranean mussel *Mytilus galloprovincialis* [40]. This ex vivo approach fulfills the 3R principle (Refinement, Reduction, and Replacement) and animal welfare regulations promoting responsible scientific experimentation [41]. Despite their validated reliability, the use of *M. galloprovincialis* PCTSs to assess the reactivity of bioactive compounds remains limited. This approach is particularly valuable within the One Health framework, which recognizes the health of ecosystems, animals, and humans as closely linked and interdependent. In this context, evaluating the effects of natural marine products in non-target invertebrate species such as mussels contributes to a more holistic understanding of both their potential benefits and ecological risks.

The potential hypolipidemic *FFB-like* effect of CAU was investigated through an integrated approach that combined measurements at both the molecular and cellular levels of responses, reflecting the activation of lipolysis, modulation of the overall lipid metabolism, and susceptibility of redox homeostasis. Parallel analyses were performed on PCTSs exposed to the reference hypolipidemic agent FFB to enable comparative evaluation.

This pioneering study was expected to provide novel insights into the cellular response mechanisms elicited by CAU in non-target species, providing valuable information that could facilitate its future valorization as a marine natural product with potential therapeutic applications.

## 2. Results

Figure 1 shows the results of lipid metabolism-related parameters in PCTSs exposed to fenofibrate (FFB) and caulerpin (CAU) for 24, 48, and 72 h.

A decreasing trend in neutral lipid content compared to the control is shown for both FFB and CAU-exposed PCTSs, with significant differences after 48 and 72 h (Figure 1A,B; Appendix A). No variation in ACOX activity is observed for both FFB and CAU at 24 and 48 h (Figure 1C,D), while a significant induction was measured for CAU after 72 h of exposure (Figure 1D). At the molecular level, a statistically significant increase in ACOX transcript levels compared to the control was observed in PCTSs treated with FFB for 48 h (Figure 1E), while no variations were detected for CAU and the other FFB exposure times (Figure 1E,F).

Figure 2 shows the results of oxidative stress-related responses in PCTSs exposed to FFB and CAU for 24, 48, and 72 h. At all exposure times, no significant differences in mRNA levels of genes related to oxidative stress (*Cu, Zn superoxide dismutase, catalase, glutathione S-transferase*, and *70 kDa heat shock proteins*) were observed in PCTSs exposed to FFB and CAU compared to controls (Figure 2A–H). However, at the historical level, a marked increase in lipofuscin content was measured in PCTSs treated with both molecules at all exposure times (Figure 2I,L; Appendix A).

Figure 3 reports the transcript levels of cytochrome P450 3A1 (*cyp3a1*) and ABCB1/P-glycoprotein (*abcb1*) measured in PCTSs exposed to FFB and CAU for 24, 48, and 72 h. A significant increase in *cyp3a1* levels was observed after 48 h in PCTSs exposed to FFB compared to the control (Figure 3A), while no statistical variations were detected in CAU-exposed PCTSs (Figure 3B). Transcript levels of *abcb1* were significantly higher in PCTSs exposed to FFB for 48 h, while similar values to control PCTSs were observed at 24 and 72 h (Figure 3C). CAU did not cause any significant modulation in the *abcb1* gene at all exposure times (Figure 3D).

## 3. Discussion

The discovery of new, effective active principles characterized by few side effects represents one of the main objectives in drug development research [2]. In this context, there is a growing interest in the properties of natural molecules derived from marine environments, as numerous studies have demonstrated their efficacy against a wide range of diseases. Among marine-derived compounds, the natural alkaloid caulerpin, isolated from the invasive green alga *Caulerpa cylindracea*, was investigated in the present study as a possible compound with hypolipidemic properties. This is supported by evidence demonstrating that CAU functions as a dual partial agonist of peroxisome proliferator-activated receptors (PPARs) α and γ, inducing activation of lipid metabolism-related genes comparable to that triggered by the synthetic drug FFB in aquatic non-target species, as initially predicted by computational modeling and subsequently validated through in vivo, ex vivo, and in vitro studies of marine fish *Diplodus sargus* [23,42]. Homologs of these nuclear receptor proteins have also been previously identified in marine bivalves [43], where they act as transcription factors regulating genes involved in cellular differentiation, development, inflammation, reproduction, and lipid metabolism [23,44,45,46].

To deepen the knowledge on CAU’s possible hypolipidemic properties, an ex vivo approach based on *M. galloprovincialis* PCTS exposure was applied in the present study. The application of this methodology agrees with the 3R principle (Refinement, Reduction, and Replacement) for the adoption of measures to improve the welfare of animals used in research; it requires a reduced number of organisms and lower amounts of exposure compounds for experimental settings, with advantages in both ethical issues and laboratory costs, thus significantly enhancing socio-economic and environmental sustainability. PCTSs from the mussel’s digestive gland represent a robust experimental model owing to the preservation of the tissue’s natural architecture, thereby maintaining physiological relevance and more accurately reflecting in vivo conditions compared to isolated cells, which frequently exhibit increased sensitivity [40]. The integrity and responsiveness of this model have been extensively validated for studies assessing molecular, biochemical, and cellular parameter variations [40]. Moreover, the use of PCTSs has a wide range of applications in biological sciences: beyond toxicological assessments, they facilitate the investigation of mechanisms of action of chemical substances and of the metabolism of pharmaceutical compounds in both target and non-target species [47,48,49,50,51,52]. However, despite their considerable potential, the use of PCTSs from marine invertebrates remains limited in studies focusing on bioactive molecules, highlighting the rationale for employing this technique in the present study using *Mytilus galloprovincialis* as a model organism.

In this respect, results obtained from PTCS exposure in mussels revealed a decrease in neutral lipid content in response to both CAU and FFB, with marked effects after 48 h and 72 h of exposure, thus highlighting, for the first time, the lipid-lowering effect of CAU in mussels. In model species, the FFB hypolipidemic function relies on its role as a PPAR agonist, and the known mechanism of action consists of the activation of lipolysis processes through the β-oxidation of long-chain fatty acids and lipid derivatives mediated by Acyl-CoA oxidase [53,54]. In the present study, the decrease in neutral lipids was parallel with the enhanced mRNA levels of ACOX in mussels treated with FFB (48 h) and the induction of ACOX enzymatic activity after CAU treatments (72 h). In addition to suggesting similarities in FFB’s and CAU’s effects, such findings revealed, for the first time, that CAU’s hypolipidemic properties might possibly be related to the activation of lipid catabolism in invertebrate marine species, an aspect so far seldom investigated. In agreement with data obtained in the present study, the CAU-mediated activation of the lipid β-oxidation pathway was also demonstrated in both ex vivo and in vivo exposures of the white seabream *D. saragus* treated with CAU [23].

In the present study, to provide a more comprehensive overview of CAU’s properties and further characterize its potential benefits and harms, the ability of CAU to alter redox homeostasis was assessed by measuring the accumulation of lipid peroxidation products and the expression of oxidative stress-related genes. At the histological level, CAU and FFB treatments highlighted, at all exposure times, the enhancement in the lipofuscin content. Lipofuscin is a heterogenous amalgam mainly composed of oxidized proteins (30 to 70%) and lipids such as triglycerides, free fatty acids, cholesterol, and lipoproteins (20 to 50%), while carbohydrates represent a small contribution that may increase proportionally with age (4 to 7%) [55]. Lipofuscin is widely regarded as a general biomarker of oxidative stress, reflecting the accumulation of oxidized macromolecules at cellular and tissue levels. Although its increase is typically associated with the onset of oxidative stress [56,57,58,59,60], our gene expression results highlighted that exposure to both CAU and FFB did not cause variations in oxidative stress-related genes, thus allowing us to hypothesize that the increase in lipofuscin levels may be related to enhanced lipid catabolism and degradation rather than to impairment of oxidative homeostasis. A similar finding, reporting a link between lipofuscin formation and the oxidation of polyunsaturated fatty acids, has previously been observed in the housefly *Musca domestica* [61].

Moreover, cellular metabolism and peroxisomal reactions induced by FFB and CAU can promote the generation of reactive oxygen species (ROS) [27,28,62]; however, under physiological conditions, ROS production is neutralized by antioxidant defenses [63,64,65]. This mechanism may explain the absence of significant variations in oxidative stress-related responses observed at the molecular level in the present study [55,66,67]. In agreement with our results, previous investigations on white seabream *D. sargus* and *M. galloprovincialis* confirmed the low prooxidant challenge of CAU [20,24]. This evidence is consistent with results from human cell lines, suggesting that CAU is also characterized by anti-inflammatory and anti-atherogenic properties [20]. On the other hand, 28-day in vivo exposure of mussels to CAU and FFB revealed that FFB markedly induced oxidative stress-related morphological alterations in the gills and digestive tubules, whereas CAU did not cause any significant changes in exposed mussels [24]. In the present study, only limited differences between CAU and FFB were observed in the responses of oxidative pathways. It is important to consider that oxidative stress responses may not necessarily correlate with transcriptional changes in genes but rather involve post-translational modifications of proteins [56]. Furthermore, several processes can mask the links between effects occurring at various intracellular levels, including different timings for transcriptional and translational mechanisms, metabolic capability of tissues, post-transcriptional modifications of proteins, and bi-phasic responses of antioxidant enzymes [56]. For example, a significant induction of catalase activity was observed in brown trout (*Salmo trutta*) exposed to heavy metals, but without variations in mRNA expression [68], suggesting a different time-course of these responses and/or a shorter half-life of mRNA [56]. Therefore, since gene modulation does not always reflect functional responses to oxidative challenge, further investigation may help avoid underestimation of the pro-oxidative effects caused by CAU and FFB by integrating gene expression data with variations in catalytic activity.

In this work, changes in the mRNA levels of *cyp3a1* and *abcb1* genes, involved in the metabolism and excretion of xenobiotic compounds, respectively, were evaluated.

The results showed an increase in the expression of both genes after 48 h exposure to FFB, while no variations were observed in CAU treatments, suggesting a different time of action between these two molecules. Given that the half-life of the synthetic drug FFB in humans is estimated to range between 19 and 27 h [69], the delayed manifestation of its effects on metabolic (*cyp3a1*) and excretory (*abcb1*) pathways, observed only after 48 h of exposure, can be adequately justified. Interestingly, although not statistically significant, the induction of *cyp3a1* in FFB-exposed PCTSs appears to remain unchanged even after 72 h of exposure, indicating ongoing biotransformation processes. However, this was accompanied by an unexpected decrease in *abcb1* mRNA levels, suggesting a reduction in FFB excretion. Although our results showed no significant changes in the expression levels of metabolism- and excretion-related genes in response to CAU exposure, other studies have demonstrated that CAU accumulation can affect key cellular processes in fish, primarily those involving the CYP450 biotransformation pathway. Increased enzymatic activities and mRNA levels of cytochrome P450 were measured in *D. saragus* sampled from sites with abundant *C. racemosa* proliferation [20], suggesting that CAU metabolism may require longer times than those applied in the present experimental design.

## 4. Conclusions

Overall, this pioneering study demonstrated the validity and applicability of PCTSs from mussels’ digestive glands as a reliable ex vivo model for assessing the effects of bioactive compounds across multiple levels of biological organization, offering a valuable platform to extend such investigations to a broader spectrum of molecules. The obtained results demonstrated the ability of the natural alkaloid CAU to reduce neutral lipid content in PCTSs of *Mytilus galloprovincialis* digestive glands. The comparison with the synthetic drug FFB provided preliminary insights into potential similarities between the two tested active principles, suggesting that CAU may act as a PPARα agonist. However, further mechanistic studies are required to confirm this hypothesis. Interestingly, CAU showed slower metabolism than FFB, and its potential hypolipidemic effects did not appear to be associated with detrimental oxidative stress responses. Nevertheless, additional investigations are necessary to clarify any potential adverse effects of CAU in marine invertebrates. In conclusion, these findings offer a preliminary but promising contribution to the understanding of caulerpin’s biological activity and support the rationale for further exploring its potential valorization as a marine-derived compound for pharmaceutical or nutraceutical applications.

## 5. Materials and Methods

### 5.1. Solutions and Culture Medium

The composition of buffers and culture medium was based on standardized protocols [40]. Physiological solution (PS) (NaCl 436 mM, KCl 10 mM, CaCl_2_ 10 mM, MgSO_4_ 53 mM; pH 7.3) and Leibovitz’s medium (L-15) (Gibco; adjusted to NaCl 436 mM, KCl 10 mM, CaCl_2_ 10 mM, MgSO_4_ 53 mM, supplemented with L-glutamine 2 mM) were sterilized and supplemented with a 1% penicillin/streptomycin antibiotic mix.

### 5.2. Animal Maintenance

*Mytilus galloprovincialis* mussels (5.5 ± 1 cm) were obtained from a local aquaculture farm (Ancona, Adriatic Sea, Italy) and acclimatized for two weeks with recirculating Artificial Sea Water (ASW) at 35 ‰ salinity and 20 °C (2 mussels/L). The animals were fed daily with a commercial mixture of marine plankton (EasySPS EVO, size range 0.2–400 μm), and the water was changed three times a week.

### 5.3. Exposure of Precision-Cut Tissue Slices (PCTSs)

PCTSs were prepared from digestive glands (DGs) according to a standardized method [43]. Briefly, each DG was excised carefully, the gastro-intestinal tract was washed with cold sterilized ASW, and DGs were maintained in cold (4 °C) sterilized ASW until slicing. DGs were included in 2.5% low-melting-point agarose and sectioned with a vibratome (VT1200S, Leica, Wetzlar, Germany) set with the following parameters: thickness: 400 µm, amplitude: 0.1 mm, and speed: 0.18 mm/s. Immediately after slicing, the PCTSs were placed in cold PS supplemented with 10 mM D-glucose and then pre-incubated in a 12-well plate (4 PCTSs/well) in L-15 with 10% fetal bovine serum (FBS) (1 mL/well), at 18 °C and with mild shaking (50 rpm) for 1:30 h. A total of 24 PCTSs were obtained from each DG. The exposure concentration of the tested compounds was selected based on previous findings from in vitro screening [70], which demonstrated a lack of CAU cytotoxicity under a wide range of concentrations (12.5 µM to 250 µM), along with in silico modeling and in vitro, ex vivo, and in vivo assays [23], which showed the activation of PPARα- and PPARγ-mediated transcription at CAU concentrations ≥ 10 µM. Based on this background information, preliminary dose-dependent screening (10, 20, 50, and 100 µM) was conducted, and 100 µM was identified as the most suitable concentration for eliciting measurable molecular, biochemical, and cellular effects in PCTSs of mussels’ digestive glands. Full details on active principles are given in the Appendix A.

After pre-incubation, PCTSs were exposed to FFB 100 µM (36.08 mg/L) and CAU 100 µM (39.84 mg/L) in L-15 medium (both dissolved in DMSO at 1% final concentration) for 24 h, 48 h, and 72 h. A control (PCTSs in L-15 alone) and a vehicle control (PCTSs in L-15 with 1% DMSO) were included. For both FFB and CAU exposure, the PCTSs from each individual DG (24 PCTSs/DG) were distributed in 6 wells of a 12-well plate (4 PCTSs/well) and incubated in 1 mL of control, vehicle control, or exposure medium (8 PCTSs/condition). Every 24 h, the culture medium was renewed, and FFB or CAU was re-dosed. The experimental plan was replicated many times to obtain, at the end of the exposure, for each treatment, 2 PCTSs to include in cryostat embedding medium (Killik, Bio Optica, Milan, Italy), frozen at −80 °C for histological analysis; 6 PCTSs to pool and frozen at −80 °C for each gene expression analysis; and 16 PCTSs (pooled from 2 different DGs) frozen at −80 °C to evaluate the Acyl-CoA oxidase’s (ACOX) activity. The viability and the tissue integrity of PCTSs were evaluated and confirmed over the 72 h exposure period using the Alamar Blue assay, a resazurin-based method (TOX-8, Sigma-Aldrich, Saint Louis, Missouri, USA), and hematoxylin-eosin staining, respectively.

### 5.4. Histological Analysis of Neutral Lipids and Lipofuscin

Neutral lipid and lipofuscin staining was performed on 4 cryostat sections (10 µm) obtained from 2 PCTSs (400 µm) from the 3 DGs used for each treatment and experimental time. Cryosections were fixed in Beker’s fixative (2.5% NaCl) for 15 min and stained both using Oil Red O (ORO) before mounting in glycerol gelatin and via the Schmorl reaction before mounting with Eukitt to determine neutral lipid and lipofuscin content, respectively. Slides were observed under a light microscope. For both analyses, four measurements were made on digestive tubules of each section (one section from each of the 2 PCTSs of the 3 DGs for each treatment). Quantification of staining intensity was performed with Image-Pro^®^ Plus 6.2 Analysis Software and then normalized to the area of digestive tubules. The results were expressed as the fold change in the control and averaged (n = 3).

### 5.5. Quantitative Analysis of mRNA Levels

mRNA levels of acyl CoA oxidase 1 (*acox1*), catalase (*cat*), glutathione S-transferase pi-isoform (*gst-pi*), 70 kDa heat shock proteins (*hsp70*), Cu, Zn superoxide dismutase (*Cu/Zn-sod*), cytochrome P450 3A1 (*cyp3A1*), and ATP-binding cassette (ABC) transporter (*abcb1*) were quantified through absolute real-time PCR (qPCR).

Total RNAs were extracted from 4 pools of PCTSs from 4 mussels (each pool was composed of 6 PCTSs from a single digestive gland) for each treatment and experimental time, using the Hybrid-R kit (GeneAll, Seoul, Korea) according to the manufacturer’s protocol. Total RNA concentrations and purity were measured using spectrophotometric methods (NanoDrop, ND1000, EuroClone, Milan, Italy); 260/280 and 260/230 ratios were >1.8. cDNA samples were synthesized from 1 µg of total RNA through retrotranscription reactions using oligo-dT and random examers (iScript cDNA synthesis, BioRad, Hercules, USA). qPCRs were performed with the Sybr Green method in a 96-well thermocycler (StepOnePlus, Applied Biosystems, Life Technologies, Basel, Switzerland). Reactions were set up with 7.5 μL of SYBR Select Master Mix (Life Technologies, Basel, Switzerland), 5 μL of total cDNA (diluted 1:5), and 0.2 mM of forward and reverse primers (Appendix A). Appendix A indicates the annealing temperatures for all genes. A non-template control was included. For each gene, a standard curve was built with 8 serial dilutions of a plasmid at known copy numbers containing the amplicon of interest. Samples and standards were amplified in duplicate in the same run. A calibration curve was built by plotting cycle threshold (Ct) values of standards versus log_10_ copy numbers. The Ct values resulting from each sample run were interpolated on the standard curve and converted into copy number/µg of total RNA. The results were expressed as the fold change in the control and averaged (n = 4).

### 5.6. Activity of Acyl-CoA Oxidase (ACOX)

For the evaluation of the Acyl-CoA oxidase’s (ACOX) activity, 3 pools of PCTSs from 6 mussels (each pool was composed of 14/16 PCTSs from two digestive glands) for each treatment and experimental time were homogenized in 1 mM sodium bicarbonate buffer (pH 7.6), containing 1 mM EDTA, 0.1% ethanol, and 0.01% Triton X-100, and centrifuged at 500× *g* for 15 min at 4 °C. H_2_O_2_ production was measured using a coupled assay [71] by following the oxidation of dichlorofluorescein diacetate (DCF-DA) catalyzed by exogenous horseradish peroxidase (HRP). The reaction medium was 0.5 M potassium phosphate buffer (pH 7.4), 2.2 mM DCF-DA, 40 μM sodium azide, 0.01% Triton X-100, and 1.2 U/mL HRP in a final volume of 1 mL. After pre-incubation at 25 °C for 5 min in the dark with an appropriate amount of sample, reactions were started by adding the substrate Palmitoyl-CoA at final concentrations of 30 μM and 100 μM for Acyl-CoA oxidase (ACOX); readings were carried out at λ = 502 nm against a blank without the substrate. The results were expressed as the fold change in the control (n = 3).

### 5.7. Statistical Analysis

Statistical analyses were performed using RStudio (version 2023.12.0 Build 369). Data were assessed for normal distribution using the Shapiro–Wilk test and for homogeneity of variances using Levene’s test. For each exposure time, the comparison between the treatment and its solvent control was evaluated using one-way analysis of variance (ANOVA), with the level of significance at the 95% confidence interval, α = 0.05. Post hoc analysis was performed using the Student–Newman–Keuls test to compare each experimental group.

## Figures and Tables

**Figure 1 toxins-17-00512-f001:**
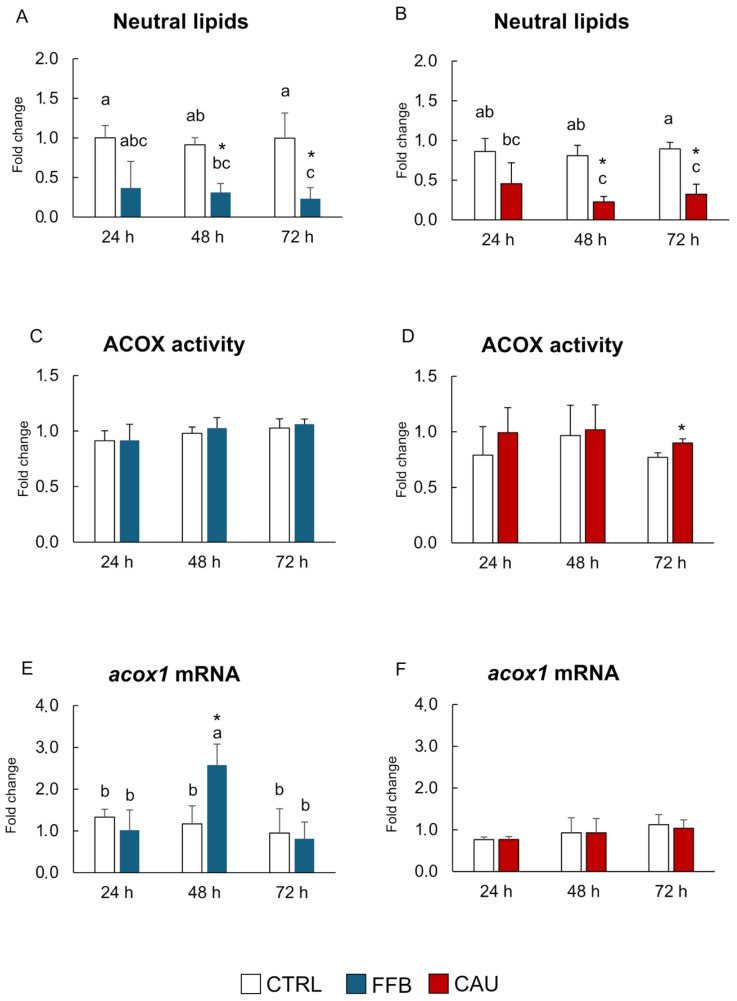
Lipid metabolism-related parameters. Content of neutral lipids (**A**,**B**), catalytic activities (**C**,**D**), and mRNA levels (**E**,**F**) of acyl CoA oxidase (ACOX) in PCTSs exposed, for 24, 48, and 72 h, to CTRL in DMSO (white), FFB (blue), and CAU (red). Data are expressed as mean values ± standard errors (n = 3). * (*p*-value < 0.05) represents statistical significance compared to solvent control PCTSs, determined using one-way ANOVA. Bars sharing the same letter are not significantly different (*p* < 0.05; results of post hoc Student–Newman–Keuls test). DMSO, dimethyl sulfoxide; CTRL, control; FFB, fenofibrate; CAU, caulerpin.

**Figure 2 toxins-17-00512-f002:**
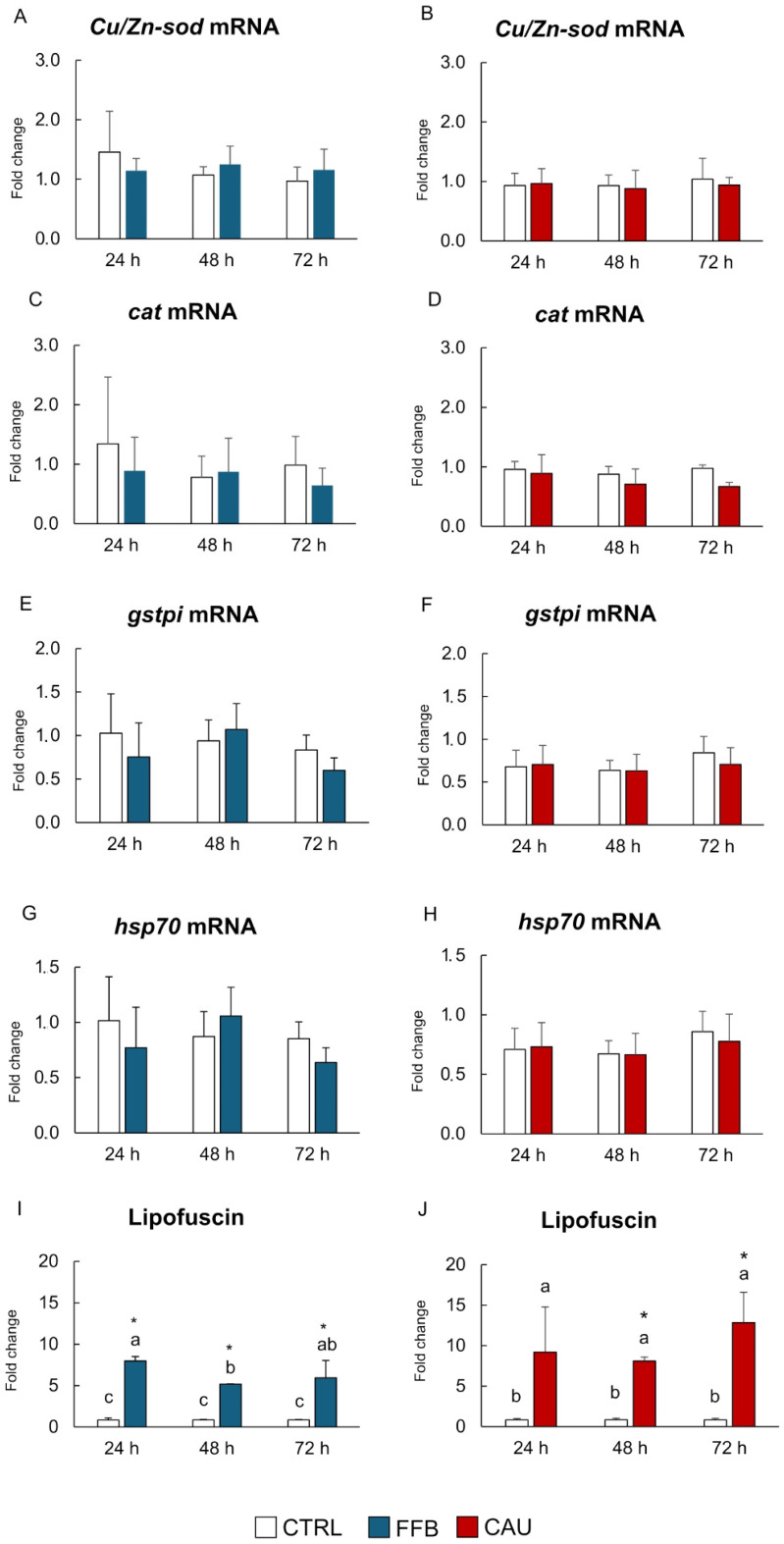
Oxidative stress-related responses. mRNA levels of *Cu/Zn-sod* (**A**,**B**), *cat* (**C**,**D**), *gst-pi* (**E**,**F**), and *hsp70* (**G**,**H**); content of lipofuscin (**I**,**J**) in PCTSs exposed, for 24, 48, and 72 h, to CTRL in DMSO (white), FFB (blue), and CAU (red). Data are expressed as mean values ± standard errors (n = 3). * (*p*-value < 0.05) represents statistical significance compared to solvent control PCTSs, determined using one-way ANOVA. Bars sharing the same letter are not significantly different (*p* < 0.05; results of post hoc Student–Newman–Keuls test). DMSO, dimethyl sulfoxide; CTRL, control; FFB, fenofibrate; CAU, caulerpin.

**Figure 3 toxins-17-00512-f003:**
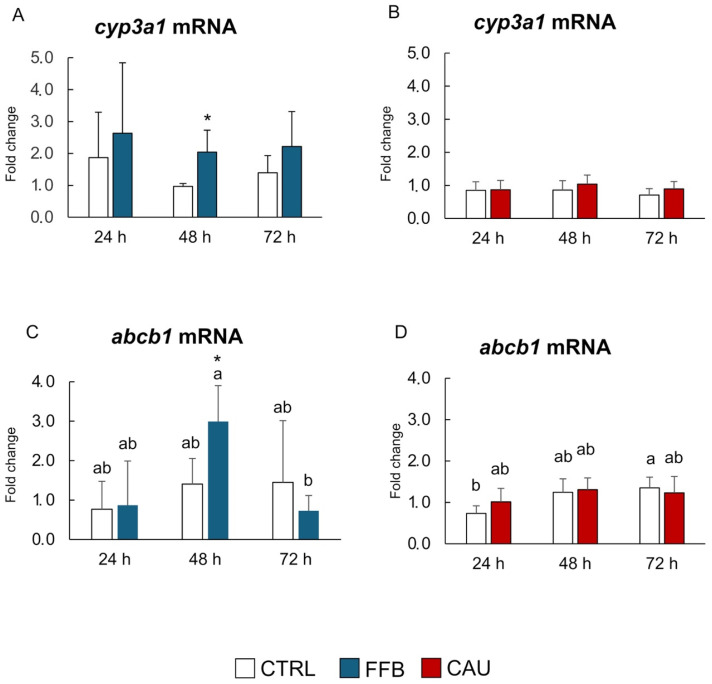
Metabolism/excretion-related genes. mRNA levels of *cyp3a1* (**A**,**B**) and *abcb1* (**C**,**D**) in PCTSs exposed, for 24, 48, and 72 h, to CTRL in DMSO (white), FFB (blue), and CAU (red). Data are expressed as mean values ± standard errors (n = 3). * (*p*-value < 0.05) represents statistical significance compared to solvent control PCTSs, determined using one-way ANOVA. Bars sharing the same letter are not significantly different (*p* < 0.05; results of post hoc Student–Newman–Keuls test). DMSO, dimethyl sulfoxide; CTRL, control; FFB, fenofibrate; CAU, caulerpin.

## Data Availability

The original contributions presented in this study are included in the article/Appendix A. Further inquiries can be directed to the corresponding authors.

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
