# Peer review of "Mechanisms of Cellular Responses of the Natural Alkaloid Caulerpin and Its Similarities with the Lipid-Lowering Agent Fenofibrate in Mytilus galloprovincialis"

_toxins, 2025, doi:10.3390/toxins17100512_

Round 1

Reviewer 1 Report

Comments and Suggestions for Authors

Major revisions:

1) Although the Authors rightly claim to have used PCTS from the digestive gland (DG) of Mytilus galloprovincialis by the ethical principles of the National Centre for the Replacement, Refinement, and Reduction of Animals in Research (NC3Rs),  the results obtained in Mytilus galloprovincialis, an invertebrate, cannot be extended to vertebrates to propose caulerpin as a potential lipid-lowering nutraceutical in humans. Therefore, I strongly suggest the Authors extend their analyses to PCTS (precisely cut intestinal slices) from vertebrates such as pigs, sheep, or cattle for their tests. These slices can be easily obtained from municipal/provincial slaughterhouses or, even better, from human intestinal resections obtained during surgeries, thus avoiding the unnecessary killing of animals. In this case, the results would be much more reliable and applicable to humans.

2) To analyse oxidative stress, the Authors should test not only the levels of detoxifying enzymes, but also the production of ROS, for example, by measuring the presence of oxidised proteins in PCTS using immunohistochemical staining of aminoacylcarbonyls.

3) The Authors never indicate whether the viability and morphological integrity of the DG slices were assessed, for example, by measuring ATP concentrations and by histological analysis using hematoxylin-eosin staining.

Author Response

Comment 1: Although the Authors rightly claim to have used PCTS from the digestive gland (DG) of Mytilus galloprovincialis by the ethical principles of the National Centre for the Replacement, Refinement, and Reduction of Animals in Research (NC3Rs), the results obtained in Mytilus galloprovincialis, an invertebrate, cannot be extended to vertebrates to propose caulerpin as a potential lipid-lowering nutraceutical in humans. Therefore, I strongly suggest the Authors extend their analyses to PCTS (precisely cut intestinal slices) from vertebrates such as pigs, sheep, or cattle for their tests. These slices can be easily obtained from municipal/provincial slaughterhouses or, even better, from human intestinal resections obtained during surgeries, thus avoiding the unnecessary killing of animals. In this case, the results would be much more reliable and applicable to humans.

Answer 1: We would like to thank the Reviewer for these thoughtful and constructive comments. We would like to emphasize that the main objective of the present study was not to propose caulerpin as a lipid-lowering nutraceutical for human use, but rather to assess its capacity to modulate lipid metabolism in Mytilus galloprovincialis. In doing so, we have also explored the potential of this marine-derived compound as a candidate for future investigations, including studies in higher organisms, should its efficacy and safety be further supported. Mytilus galloprovincialis was selected as an experimental model because it has been extensively validated in ecotoxicology research, and it is well established that its digestive gland elicits several metabolic functions analogous to those of the vertebrate liver and gut. Notably, the presence of PPAR homologs in Mytilus and other mollusks further supports its relevance for studying molecular interactions involving bioactive molecules such as caulerpin and fenofibrate. As also kindly defined by #Reviewer 3, this study represents a "pioneering investigation" using a low-impact, ethically sustainable invertebrate model, to study the potential lipids-regulatory effects of caulerpin. Additionally, we believe that applying the 3Rs principles (Replacement, Reduction, and Refinement) to invertebrate research is increasingly important. As ethical standards evolve, future regulations may extend to cover the welfare of invertebrates in in vivo experimentation. Although we agree with the reviewer that the use of vertebrate models or human tissues would certainly provide results with greater translational potential, the development and validation of alternative invertebrate models now may contribute to reducing the use of vertebrates and enhancing animal welfare in scientific research overall. The approach used in the present study is aligned to the One-Health perspective that recognizes the health of ecosystems, animals and humans as closely linked an interdependent. All these aspects have now been included in the revised version of the present MS (Introduction, LINES: 81-87). We appreciate having the opportunity to clarify the aim and scope of our study (LINES: 71-97) as also suggested by #Reviewer 2.

Comment 2: To analyse oxidative stress, the Authors should test not only the levels of detoxifying enzymes, but also the production of ROS, for example, by measuring the presence of oxidised proteins in PCTS using immunohistochemical staining of aminoacylcarbonyls.

Answer 2: We are aware that oxidative stress is a critical condition derived from the imbalance between the generation of reactive oxygen species and the sophisticated network of antioxidant mechanisms. In this respect, in the present study, the levels of detoxifying enzymes were carried out along with measurements of lipofuscin levels, a more general marker of lipid peroxidation induced by ROS. Indeed, although lipofuscin composition may vary across different tissues, it is highly recognized by the scientific community that in most cases it is a complex mixture containing mainly oxidized, cross-linked proteins, lipids, and only minor amounts of carbohydrates (Baldensperger et al., 2024). Its accumulation is widely used as a proxy for cumulative oxidative damage, particularly in long-lived cells and tissues with high metabolic activity, such as the digestive gland in Mytilus galloprovincialis. We fully agree that additional oxidative stress endpoints, such as the immunohistochemical detection of aminoacylcarbonyls, could provide valuable insights and we will certainly consider including this more specific marker in future studies. All these aspects have now been included in the revised version of the discussion section (LINES: 202-213).

Comment 3: The Authors never indicate whether the viability and morphological integrity of the DG slices were assessed, for example, by measuring ATP concentrations and by histological analysis using hematoxylin-eosin staining.

Answer 3: We thank the Reviewer for this important observation and for the opportunity to strengthen the methodological clarity. Indeed, we would like to specify that all aspects related to the viability and morphological integrity of the digestive gland (DG) slices, which form the essential basis for the validation of PCTS in laboratory experiments, have already been thoroughly addressed and validated in Giuliani et al., 2019 (ref. already included in the paper). That study provided comprehensive testing of viability and structural integrity parameters for PCTS of Mytilus galloprovincialis digestive glands, supporting the robustness and applicability of this model in toxicological research. In our specific case, viability and morphological integrity were assessed using two complementary approaches: (i) the Alamar Blue assay, a resazurin-based method (TOX-8, Sigma-Aldrich) to evaluate cell metabolic activity, and (ii) histological analysis with hematoxylin-eosin staining, to verify tissue structure. These evaluations confirmed that the slices remained viable and morphologically intact throughout the entire 72-hour exposure period. We have now clarified this point more explicitly in the revised manuscript (LINES: 77-79; 302-305). 

Reviewer 2 Report

Comments and Suggestions for Authors

The submitted manuscript need major revision before consideration;

  1. first of all the title is very strange it should be specific and attracted to the wide readers of this journal
  2. The abstract has many unwanted text such as start of the abstract (A growing interest arose toward pharmacological properties of the largely un- 6
    explored marine-derived natural products, considered to have a great potential against a wide array of diseases. Cardiovascular diseases represent one of the major health burdens of the modern era, closely linked to elevated plasma lipid levels and typically treated with synthetic pharmaceuticals such as statins or fibrates )as well as "alkaloid caulerpin, a secondary metabolite of green algae of the genus Caulerpa"
  3. Authors should more detail about the objective of the study.
  4. The contents and quantity of the manuscript is not soo impressive, there should more drugs either natural or synthetic to compare
  5. Authors have not given the structure of the alkaloids as well as synthetic drug  
  6. Either there is any SAR in the manuscript for the more precise of the results
  7. Author can draw the comparative study on structural base
  8. The presentation of the results is not convincing it should be more comprehensive
  9. Authors should give the conclusion of the study

Author Response

Comment 1: First of all the title is very strange it should be specific and attracted to the wide readers of this journal

Answer 1: Considering this comment, we decided to revise the title to make it more specific and appealing to a broader readership, in line with the scope of the journal: “Mechanisms of cellular responses of the natural alkaloid caulerpin and its similarities with the lipid lowering agent fenofibrate in Mytilus galloprovincialis”

Comment 2: The abstract has many unwanted text such as start of the abstract (A growing interest arose toward pharmacological properties of the largely un- 6
explored marine-derived natural products, considered to have a great potential against a wide array of diseases. Cardiovascular diseases represent one of the major health burdens of the modern era, closely linked to elevated plasma lipid levels and typically treated with synthetic pharmaceuticals such as statins or fibrates) as well as "alkaloid caulerpin, a secondary metabolite of green algae of the genus Caulerpa"

Answer 2: We agree with the reviewer’s observation. The Abstract has now been carefully revised by removing redundant and non-essential information, in order to improve clarity and focus. 

Comment 3: Authors should more detail about the objective of the study.

Answer 3: The study objective has now been revised to enhance clarity and precision, also following #Reviewer 1 feedback (LINES: 71-97).

Comment 4: The contents and quantity of the manuscript is not soo impressive, there should more drugs either natural or synthetic to compare.

Answer 4: We thank the Reviewer for this observation. As highlighted by #Reviewer 3, this is a "pioneering study" that aims to explore the feasibility and preliminary effects of the selected compounds in the selected ex vivo model. Expanding the number of drugs tested, whether natural or synthetic, would indeed represent an important next step. However, such an investment (both in terms of resources and experimental complexity) can be realistically justified only once the efficacy of the proposed approach is further supported by early evidence. The encouraging results obtained so far point in this direction and lay the foundation for future studies. These will allow us to build a more robust dataset, expand the panel of tested molecules, and include a broader range of analytical parameters. All these aspects have now been addressed in the revised version of the manuscript (LINES 81-83; 180-183; 362-366).

Comment 5: Authors have not given the structure of the alkaloids as well as synthetic drug  

Answer 5: We thank the Reviewer for this important comment. The chemical structures of caulerpin and fenofibrate have now been added as a figure in the revised version of Supplementary Materials.

Comment 6-7-8: Either there is any SAR in the manuscript for the more precise of the results; Author can draw the comparative study on structural base; The presentation of the results is not convincing it should be more comprehensive

Answer 6-7-8: We thank the reviewer for these three comments, which are closely interconnected, and we are pleased to address them together in a unified response.While our current work did not include a formal Structure–Activity Relationship (SAR) analysis, previous research by Vitale et al. (2019) has extensively characterized the interaction of caulerpin (CAU) with PPAR receptors using a multidisciplinary approach, including in-silico modeling, in-vitro, ex-vivo, and in-vivo assays in another marine species (D. sargus). Obtained results demonstrated that CAU acts as a dual partial agonist of PPARα/γ. Noteworthy, the CAU-binding modes were shown to involve hydrophobic interactions distinct from those classical full agonists, and the activation of lipid metabolism-related genes comparable to that induced by FFB has also been observed. This evidence guided our choice to use FFB as a reference compound for comparative purposes and provided a structural and functional framework supporting the pharmacological potential of CAU. Additionally, the recent review by D’Aniello et al. (2023) further corroborates the partial agonist behaviour of CAU on PPAR isoforms, highlighting molecular binding similarities with other known ligands and its metabolic implications in marine species. In light of these prior findings, the MS has now been revised (LINES: 155-160) to better contextualize our findings within this structural and mechanistic background and to strengthen the rationale behind the comparison between CAU and FFB in our experimental model, thus contributing to a more comprehensive and convincing presentation of the data.

Comment 9: Authors should give the conclusion of the study

Answer 9: We thank the Reviewer for the helpful suggestion. We agree that a conclusion section improves the clarity and completeness of the manuscript. Accordingly, we have added a dedicated Conclusions section at the end of the manuscript (LINES: 362-376)

Reviewer 3 Report

Comments and Suggestions for Authors

The study presents pioneering evidence on the hypolipidemic potential of the marine-derived alkaloid caulerpin, contributing valuable insight into the pharmacological utility of marine natural products. The parallel investigation of caulerpin and fenofibrate strengthens the study’s design, allowing a meaningful comparison between a natural compound and a clinically used lipid-lowering agent. However, several clarifications are needed.

Major comments

Suggest to add a brief mention of the extent of lipid reduction or % changes could help quantify the significance of the findings in the abstract section.

The study only evaluates a single concentration 100 µM of caulerpin and fenofibrate. Dose-dependency is a critical parameter to establish pharmacological relevance and potential toxicity thresholds.

There is an apparent contradiction where lipofuscin accumulation is observed, but no changes are seen in antioxidant gene expression.

Please mention what a, b, ab, and abc denote in figures 1, 2, and 3.

This manuscript contains incorrect prepositions, passive phrasing, missing hyphens, and verb usage inconsistencies. This should be revised carefully.

Author Response

Comment 1: The study presents pioneering evidence on the hypolipidemic potential of the marine-derived alkaloid caulerpin, contributing valuable insight into the pharmacological utility of marine natural products. The parallel investigation of caulerpin and fenofibrate strengthens the study’s design, allowing a meaningful comparison between a natural compound and a clinically used lipid-lowering agent. However, several clarifications are needed.

Answer 1: We sincerely thank the reviewer for the positive and encouraging comments. We truly appreciate the recognition of the innovative nature of our study and the relevance of comparing a marine-derived compound with an established lipid-lowering agent. We have carefully addressed the points requiring clarification, which helped improve the overall quality and clarity of the manuscript.

Comment 2: Suggest to add a brief mention of the extent of lipid reduction or % changes could help quantify the significance of the findings in the abstract section.

Answer 2: We thank the Reviewer for the valuable suggestion. We have added a brief mention of the percentage reduction in lipid content observed in PCTS exposed to both compounds in the Abstract (LINES: 12-14) to better quantify the significance of our findings.

Comment 3: The study only evaluates a single concentration 100 µM of caulerpin and fenofibrate. Dose-dependency is a critical parameter to establish pharmacological relevance and potential toxicity thresholds.

Answer 3: We thank the Reviewer for this important observation. We fully agree that the evaluation of dose-dependent effects is important to better define the biological reactivity of tested compounds. In the current study, the selection of the exposure dose was based on previous findings obtained from in vitro screening (Rocha et al., 2007) demonstrating the lack of caulerpin cytotoxicity under a wide range of concentrations (from 12.5 µM up to 250 µM). Moreover, Vitale et al. (2019) demonstrated the activation of PPARα- and PPARγ-mediated transcription following exposure to CAU at concentrations higher than 10 µM. Building on this background information, we conducted preliminary dose-dependent screening using 10, 20, 50, and 100 µM concentrations, in our experimental model. A description of this experiment has now been added to the Materials and Methods section (lines 282–290). These trials indicated that 100 µM was the most suitable concentration for eliciting measurable molecular, biochemical, and cellular effects in our experimental model, PCTS of mussels' digestive glands. As also stated by the same Reviewer (#3), this study was intended as a preliminary investigation aimed at characterizing the effects of CAU and we therefore focused on a single, pharmacologically relevant dose to minimize complexity at this initial stage. The encouraging results obtained so far will certainly guide and support future investigations aimed at defining a detailed dose–response profile of tested molecules.

Comment 4: There is an apparent contradiction where lipofuscin accumulation is observed, but no changes are seen in antioxidant gene expression.

Answer 4: We get the point of the reviewer. Discrepancies in the pattern of variation of oxidative-related biomarkers measured at different levels of biological organization are not surprising, and several observations highlighted that these responses do not vary in a synchronous way, and have different time-courses of activation at the transcriptional/molecular, and functional/cellular levels: is thus quite common observing significant variation at cellular levels (e.g. accumulation of lipofuscin) while mRNA levels do not vary, or vice-versa, further highlighting the need of caution when interpreting these results in term of functional effects. This phenomenon, extensively described in the literature (e.g., Regoli and Giuliani, 2014), and often reported after laboratory exposures is consistent with our findings on significant changes in measured histopathological marker and the lack of modulation of mRNA levels. These aspects have now been better addressed in the Discussion section (LINES: 207-241) to clarify the apparent discrepancy between lipofuscin accumulation and antioxidants genes expression in our results.

Comment 5: Please mention what a, b, ab, and abc denote in figures 1, 2, and 3.

Answer 5: Ok, done. LINES: (130-131; 138-139; 145-146)

Comment 6: This manuscript contains incorrect prepositions, passive phrasing, missing hyphens, and verb usage inconsistencies. This should be revised carefully.

Answer 6: Thank you very much for your valuable and detailed comments. We have carefully revised the manuscript, correcting the incorrect prepositions, adjusting passive phrasing where appropriate, adding missing hyphens, and ensuring consistency in verb usage as per your suggestions

Round 2

Reviewer 1 Report

Comments and Suggestions for Authors

The Authors have responded exhaustively to the comments made and therefore their research can be published.

Reviewer 2 Report

Comments and Suggestions for Authors

May be accepted 

Reviewer 3 Report

Comments and Suggestions for Authors

Accept